# Mandarin classifier systems optimize to accommodate communicative pressures

**Yamei Wang and Géraldine Walther**
George Mason University
{ywang78, gwalthe}@gmu.edu

## Abstract

Previous work on noun classification implies that gender systems are inherently optimized to accommodate communicative pressures on human language learning and processing (Dye et al., 2018). The authors state that languages make use of either grammatical (e.g., gender) or probabilistic (pre-nominal modifiers) to smoothe the entropy of nouns in context. We show that even languages that are considered genderless, like Mandarin Chinese, possess a noun classification device that plays the same functional role as gender markers. Based on close to 1M Mandarin noun phrases extracted from the Leipzig Corpora Collection (Goldhahn et al., 2012) and their corresponding fastText embeddings (Bojanowski et al., 2017), we show that noun-classifier combinations are sensitive to same frequency, similarity, and co-occurrence interactions that structure gender systems. We also present the first study of the interaction between grammatical and probabilistic noun classification.

## 1 Introduction

The world's languages make use of a variety of strategies to classify nouns into sub-classes. While gendered languages like Spanish or German typically split nouns across two (masculine M vs. feminine F) or three (M, F and neuter N) classes, other languages, e.g., in the Bantu family, may split nouns across more than 10 different classes (Corbett et al., 1991; Creissels, 2006). The recurring fascination with gender systems that can be found in the linguistic literature lies in the perceived arbitrariness of gender classes, which, among other things, makes them notoriously difficult to acquire for second language learners: for example, in French or Spanish the word for sun (fr: 'soleil', es: 'sol') will be M and the word for moon (fr: 'lune', es: 'luna') will be F, while in German the genders are reversed ('Sonne' F, 'Mond' M) and in Russian 'солнце' *sólnce* sun would be N and moon

'луна' *luná* F. Yet, gender systems do not appear to present any particular challenges in first language acquisition (Arnon and Ramscar, 2012), indicating that gender systems are in fact, learnable.

As speakers, humans are confronted with two main challenges, which reflect what Blevins et al. (2017) refer to as 'competing communicative pressures': the first is learning their language, for which regularities across similar words (or structures) are the most optimal principle of organization; the second is using their language to efficiently communicate with others, for which enhanced discriminability of words and structures provides the most optimal principle. Irregularities make words and structures more item specific and thus increase their discriminability – which allows for faster and more efficient communication. A fully regular system would not be optimized for communication, whereas a fully irregular one would make a language extremely difficult if not impossible to learn. However, not all parts of the system present the same challenge for learning: Highly frequent words and structures are typically encountered early by learners and constantly reinforced. Once they are acquired, learning is no longer necessary. Irregularities in early acquired words only present a minor challenge towards the beginning of human language development, but do not pose any problems later in life. Their discriminative properties do however provide significant benefits for language processing, making communication faster and more efficient. Fast discrimination is especially useful for words and structures that speakers use most frequently. Infrequent words and structures, on the other hand, are typically only encountered late in life, if at all. Even proficient adult speakers need to leverage their knowledge about related words and structures when they first encounter them. Regularity across similar structures is essential for low frequency words and structures. At the same time,

faster processing of rarer structures is not as crucial to increasing overall communicative efficiency.

Recent work by Williams et al. (2019) showed that gender assignment tends not to be as arbitrary as it would appear, but instead correlates with nouns' semantics. Earlier work by Dye (2017) and Dye et al. (2017b, 2018) had also already shown that gender systems are inherently optimized to accommodate communicative pressures on human language learning and processing via interactions between semantic similarity, frequency, and co-occurrence likelihood: since across all languages, nouns tend to be the syntactic category with the most diverse number of items (Petrov et al., 2011), syntactic processing of nouns tends to correspond to a locus of heightened cognitive effort. This is also reflected in the fact that nouns are the most frequent sites for disfluencies, incorrect retrieval, and mishearings (Clark and Wasow, 1998; Vigliocco et al., 1997). Gender classes that make use of gendered determiners reduce the uncertainty about upcoming nouns in context by restricting the subset of possible nouns to the ones that are consistent with the gender indicated by the determiner. Dye et al. (2017b) show that the existence of interactions between nouns' semantics, their frequencies, and their assignment to gender classes closer reflects a cognitively optimized system than an arbitrary one. Dye et al.'s 2017a paper 'Cute Little Puppies and Nice Cold Beers: An Information Theoretic Analysis of Prenominal Adjectives' further illustrates how languages without gender markers, such as English, make use of a probabilistic counterpart to grammatical noun classification, where a larger number of pre-nominal modifiers progressively reduces the uncertainty of nouns in context. According to the authors, languages can make use of either grammatical (e.g., gender) or probabilisitic (pre-nominal modifier sequences) to smoothe the uncertainty (entropy (Shannon, 1948)) of nouns in context.

Our study focuses on another, different, noun classification system which is based on NOMINAL CLASSIFIERS. It relates its structure to its cognitive processability and learnability. As a grammatical category, classifiers constitute one of the most prominent features of East and South East Asian languages, such as Mandarin (Li, 2013). Classifiers categorize nouns based on distinctive semantic properties such as shape, animacy, humanness, among others (Aikhenvald and Mihas, 2019). For example, classifiers, such as 张 *zhāng* and 条 *tiáo*, are typically associated with different types of referents; 张 *zhāng* with wide and flat items like beds, nets, and maps, and 条 *tiáo* with long and narrow objects like rivers, scarfs, and legs. Yet, just like gendered determiners, classifiers can also be seen as a grammatical means for classifying nouns into different noun classes: Mandarin classifiers are part of the noun phrase. They typically appear before nouns. According to Li and Thompson (1989), they are obligatorily present after numerals or demonstratives: for instance, in phrases like 两 *liǎng* *(张 *zhāng*) 桌子 *zhuōzi* 'two tables' and 这 *zhè* *(条 *tiáo*) 椅子 *yǐzi* 'this bench', the classifiers (highlighted in bold) cannot be omitted. Nouns will only combine with certain subsets of classifiers, but not others. Just like gender markers, combining a noun with the wrong classifier would be perceived as ungrammatical.

Since knowing a noun's classifier appears just as crucial to correctly speaking Mandarin as knowing a noun's gender would be required for German or Spanish, previous studies have investigated the relation between a noun's meaning and the classifiers it combines with. Liu et al. (2019) show that classifiers cannot be fully predicted from the semantics of their nouns. Diverse nouns such as benches, smoke, passwords, and even abstract entities like regulations can all be quantified with the same classifier 条 *tiáo*. However, semantically more similar nouns like bench and table do not share classifiers (条 *tiáo* for bench, but 张 *zhāng* for table). The challenges they present for second language learners are thus very similar to those encountered by learners of gender systems (Liang, 2008). On a functional level, Wang and Walther (2023) show that classifiers also substantially reduce the conditional entropy of upcoming nouns in Mandarin processing. This indicates that knowledge of a pre-nominal classifier can help speakers predict upcoming nouns. Lau and Grüter (2015) and Grüter et al. (2020) also report that speakers use classifier-referent associations to predict upcoming nouns in Mandarin, thus facilitating noun processing.

While classifiers appear to share a great number of properties with gendered determiners, they also differ from gender markers in a number of significant aspects: (i) they do not operate over a one-to-one correspondence with specific nouns: a given noun will typically combine with multiple classifiers, of which subsets are in turn also combined

with other subsets of nouns (examples of those associations are given in Table 1); (ii) classifiers are significantly more numerous: Mandarin, for example, has hundreds of different classifiers; (iii) classifiers can sometimes appear grammatically more ambiguous since the same token will sometimes be used as a noun and sometimes as a classifier; partly as a consequence of ii (i.e., the greater diversity of classifiers) and iii (which includes the fact that classifiers can sometimes retain part of the meaning of the corresponding noun), (iv) classifiers are inherently more semantically transparent than their counterparts in gender systems.

The goal of our study is to investigate the structure and cognitive function of classifier systems compared to better known gender systems. In particular, we seek to understand how a system that is neither fully categorical (like gender systems where a noun is only ever assigned one gender) nor fully probabilistic (adjective-noun combinations of which none would be strictly ungrammatical) handles noun class membership and makes use of classifier-noun combinatorics to accommodate competing communicative pressures of learning and processing language (Blevins et al., 2017). A fully regular classifier system would translate into semantically similar nouns always sharing the same sets of classifiers. A fully irregular system would translate into semantically similar nouns never sharing the same classifiers. The hypothesis that we are testing here is whether Mandarin classifiers do indeed organize Mandarin nouns along the two competing communicative pressures related to learning and processing, which have been found to structure gender systems, and strike a balance between regularity and discriminability.

Using a large set of close to 1M Mandarin noun phrases extracted from the Leipzig Corpora Collection (Goldhahn et al., 2012) and their corresponding fastText embeddings (Bojanowski et al., 2017), we show that grammatical noun classification in the sense of Dye et al. (2017a,b, 2018) extends beyond one-to-one relations like German noungender pairs, to cases where the nominal classifiers are in a many-to-many relationship with their associated nouns. Just like gender markers, nounclassifier combinations are sensitive to noun frequencies, semantic similarity, and possible noun co-occurrence patterns. We find that semantically similar nouns indeed tend to share classifiers if they are infrequent and tend to be distributed across different classifiers when they are frequent. Additionally, we find interesting differential patterns between nouns immediately following classifiers and nouns also preceded by one or more nominal modifiers, an interaction that had never been directly investigated and reported before: classifiers will display the expected behavior of a grammatical noun class marker in the sense of Dye et al. (2017a,b, 2018) in the scenario where they immediately precede the nouns and are the only contributor to smoothing noun entropy. In cases where additional modifiers (i.e., elements providing additional probabilistic classification) are present in the noun phrase, classifier-noun pairs do not show the structure expected of grammatical noun classification. In such cases and noun classification appears to be most correlated with noun frequencies. Our study is the first to quantitatively show the effects of the co-existence of grammatical and probabilistic noun classification. In Mandarin, they manifest through differential patterns of class membership depending on both (a) noun pair similarity and (b) noun phrase structure.

## 2 Data

Our study is based on three of the 1M sentence corpora of Mandarin Chinese in the Leipzig Corpora Collection (Goldhahn et al., 2012).[1] We normalized the data by first transforming all Chinese characters into simplified Chinese utilizing the Open Chinese Convert software.[2] We then applied the CoreNLP[3] Chinese dependency parser (Chen and Manning, 2014) to our dataset. Leveraging the dependency structure, we extracted all complete nominal phrases (1,079,190 instances and 767,721 distinct phrases). Since the results of the parsing was not clean enough to conduct a detailed analysis of classifier-noun interactions, we manually filtered the extracted noun phrases as described below.

First, we removed nominal phrases with unsual lengths, specifically those exceeding 35 characters — a total of 91 noun phrases.[4] Second, any noun phrases that contained elements improperly tagged as classifiers, which included symbols, non-Chinese characters, or invalid characters,

[1]2007-2009 news, 2011 newscrawl, and 2015 China web: https://wortschatz.uni-leipzig.de/en/download/Chinese.

[2]https://github.com/BYVoid/OpenCC

[3]Version 4.4.0, released in January 2022.

[4]Extracted phrases of more than 35 characters were judged to be abnormal by one of the authors who is a native speaker.

| NPs including the same nouns | NPs including the same classifiers |
|---|---|
| 两**张**桌子 *liǎng **zhāng** zhuōzi* 'two tables' | 这**条**椅子 *zhè **tiáo** yǐzi* 'this bench' |
| 两**款**桌子 *liǎng **kuǎn** zhuōzi* 'two types of tables' | 这**条**烟 *zhè **tiáo** yān* 'this smoke' |
| 两**排**桌子 *liǎng **pái** zhuōzi* 'two rows of tables' | 这**条**规定 *zhè **tiáo** guīdìng* 'this regulation' |
| 两**车**桌子 *liǎng **chē** zhuōzi* 'two trucks of tables' | 这**条**密码 *zhè **tiáo** mìmǎ* 'this passwords' |

Table 1: Nominal phrases illustrate the multifaceted relationship between classifiers and referent nouns in Mandarin Chinese. The characters in bold are classifiers.

were excluded from our data set.[5] Lastly, nominal phrases featuring post-nominal classifiers were also omitted.[6] After data cleaning, we were left with 760,575 distinct nominal phrases that all included a pre-nominal classifier. The number of distinct nouns in our data set is 44,620.[7] Out of those, we kept 39,011 nouns occurring a minimum of 25 times to ensure that they would be represented in the pre-trained vectors used in our experiments (see section 3.1).

Since our goal included evaluating the effect of frequency on noun class membership, we divided our noun data into frequency bands of 30 and sampled 1% from each. We generated non-ordered noun pairs for those 1% and computed their summed frequency. From the pairs, we generated a data frame in which the higher and lower frequency elements of pairs were balanced[8] across first and second nouns.

Our resulting data is a set of 390 nouns, sampled from 39,011 nouns across all frequency bands to ensure that the sample reflects distributions in the full data set. From those nouns, we generated 37,831 noun pairs.[9]

## 3 Method

### 3.1 Computing the relevant measures

For each noun in our data set, we extracted the sets of classifiers it could combine with. We also computed all noun frequencies and log frequencies. We kept all nouns that occurred a minimum of 25 times in our original 3M sentence data and extracted the nouns' word vectors from the pre-trained Chinese fastText model (Bojanowski et al., 2017).[10] The vectors were included as a quantifiable measure of a noun's meaning based on the distributional hypothesis laid out by Harris (1954) and Firth (1957). We set a minimum of 25 occurrences as a threshold to ensure that all words would be represented in the set of pre-trained fastText vectors.

For each noun pair, we computed the cosine similarity between the noun vectors. As in previous studies (Dye, 2017; Dye et al., 2017b,a, 2018; Williams et al., 2019), cosine similarity was used as a measure of semantic similarity between nouns.[11]

We further extracted the co-occurrences of nouns in our corpus within a bidirectional two word window and computed their pointwise mutual information (PMI) comparing their co-occurence probability to the product of the nouns' independent probabilities, as defined by Church and Hanks (1990):[12]

$$PMI(n1, n2) = log(\frac{P(n1, n2)}{P(n1)P(n2)}) \quad (1)$$

Cosine similarity is used as a measure of gen-

---

[5]Some examples of these improperly tagged classifiers are '县、区、乡、', 'ま', and '\ue997'.

[6]This group represents approximately 0.75% of the total data and commonly occurs during item enumeration, as in the phrase 银行卡七**张** *yínhángkǎ qī zhāng* 'seven bank cards'.

[7]Given that Mandarin nouns do not inflect, we do not distinguish between lemmas and word forms.

[8]The mean frequencies and standard deviations between the first and second nouns are similar.

[9]The total of 37,831 pairs was generated under two separate conditions: 19,110 pairs were derived from 196 distinct nouns that were immediately preceded by a classifier, while 18,721 pairs were generated from 194 distinct nouns in noun phrases that also included pre-nominal modifiers.

[10]The same vectors were also used in the study by Williams et al. (2019). The High Dimensional Explorer (Shaoul and Westbury, 2010) was run to generate vectors for Dye et al. (2017b,a).

[11]In response to one reviewer's concern that (i) the presence of classifiers in the data that had been used for training the pre-trained fastText vectors might impact effects on classifier-noun combinations and (ii) the reliance of fastText on sub-word information might bias results towards form properties rather than distributional ones, we also trained customized fastText and Word2Vec (Řehůřek and Sojka, 2010) vectors on a version of our corpus from which we had previously removed all instances of classifiers. Contrarily to fastText, Word2Vec does not integrate sub-word information. Both models were trained using default settings. Overall, there were no significant differences in results involving customized vectors that would warrant a change in the experimental setup.

[12]The window size is the same as the one used by Dye et al. to evaluate corresponding effects on German noun classification based on gender.

eral semantic similarity. PMI relates to whether two nouns are likely to be used together. The PMI measure specifically targets whether speakers will be likely to encounter two nouns closely together and need to differentiate between them in context. Two words could independently have fairly similar distributions without ever actually co-occurring with one another. For example, despite their very similar meaning, 'gentleman' and 'guy' are less likely to be used together, than 'knife' and 'fork'. Words that, in addition to being semantically similar, would also be likely to co-occur are the ones for which additional disambiguation would be functionally most relevant. Introducing PMI as an additional measure allows us to test whether language systems are sensitive to co-occurrence likelihoods. Only in calculations involving PMIs, did we discard 25,073 noun pairs with negative PMIs, since they would have been computed on co-occurences too small to yield reliable information. For all other calculations the full set was used.

## 3.2 Class membership

The goal of our study is to evaluate whether Mandarin classifiers show the distribution of grammatical noun class markers defined by Dye et al. (2017b,a, 2018). The expected profile would present a correlation between same noun class membership and interactions between noun frequencies, similarities, and likelihood of co-occurrence. When evaluating gender systems, studies typically consider two nouns as belonging to the same class when they have the same gender. Since classifier systems do not display simple one-to-one correspondences, we defined a numeric same class membership score based on nouns' shared classifiers. Our metric is based on the cumulative proportion of shared classifiers across two nouns' classifier sets. For each noun pair (n1, n2), we used the classifier sets extracted from the full data for each one of our nouns to compute their shared class membership (SCM) score. SCM is defined as the sum of the number of shared classifiers across both sets (*(clf1, clf2)*) normalized on set 1 (*(clf1)*) and the number of shared classifiers (*(clf1, clf2)*) normalized on set 2 (*(clf2)*):

$$SCM(n1, n2) = \frac{(clf1, clf2)}{(clf1)} + \frac{(clf1, clf2)}{(clf2)}$$

(2)

## 3.3 Evaluating the interactions

We used a Generalized Additive Model (GAM) (Wood, 2012; Team, 2020) to evaluate the interactions between noun log frequencies, cosine similarities, and non negative PMI in predicting SCM scores as defined above.

In order to be able to evaluate two separate conditions in which (a) classifiers were the only element in a noun phrase contributing to lowering an upcoming noun's entropy and (b) classifiers and pre-nominal adjectives were jointly contributing to lowering a noun's entropy, we had to further split our data into two separate scenarios: nouns directly following classifiers, and classifier-modifier(s)-noun sequences. We generated GAM models for each one of those scenarios and for the combined data set that did not distinguish between the two.

## 3.4 Probing the effects of noun phrase structure

To further evaluate the relevance of noun phrase structure on noun-classifier combinatorics, we also computed the following measures for classifiers immediately preceding their nouns vs. classifiers separated from their nouns by additional pre-nominal modifiers:

**The mean frequency of nouns in the two scenarios:** According to Dye et al. (2018), if a noun class system makes use of probabilisitic noun classification, it should allow for lower frequency nouns to appear in noun phrases that contain additional entropy-smoothing pre-nominal modifiers. This should be reflected in the mean frequencies of nouns directly following classifiers vs. nouns in classifier-modifier(s)- noun sequences.

**The overall mutual information between classifiers and nouns:** Mutual information (MI) (Cover and Thomas, 2012) indicates how much information (in bits) is shared between a classifier and its corresponding head noun. If classifiers that immediately precede their nouns play a greater role in noun classification than those combined with probabilistic entropy smoothing, the mutual information between those classifiers and their nouns should be significantly higher. If *C* and *N* represent the sets of all classifiers and nouns respectively, and *c* and *n* their corresponding elements, then MI between each type of classifier and its corresponding noun is defined as:

$$I(N;C) = H(N) - H(N|C)$$
$$= \sum_{n \in N, c \in C} p(n,c) log \frac{p(n,c)}{p(n)p(c)} \quad (3)$$

## 4 Results

### 4.1 Frequency, similarity, and co-occurrence effects

Figures 1 and 2 show the types of interactions reported for German by Dye et al. (2018) on our full Mandarin noun data set. As can be seen from Figure 1, the likelihood of shared classifiers increases when nouns are more similar. This effect is however reduced in cases where nouns are more likely to co-occur, as indicated by a higher PMI score ($F(9.96) = 20.39$, $p < 0.0001$). This result is consistent with the results reported for German and also congruent with the study on gender classes and noun similarity by Williams et al. (2019). The frequency effects shown in Figure 2 however are the opposite of the ones found by Dye et al. for German. In our Mandarin data, likelihood of shared classifiers increases with frequency ($F(16.12) = 90.19$, $p < 0.0001$).

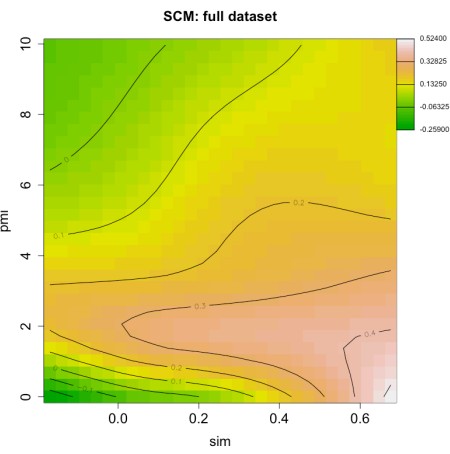

Figure 1: 2-way interaction between noun similarity and co-occurrence in full dataset: the more similar two nouns are, the more likely they are to share classifiers. This effect is however weakened for nouns occurring in similar contexts (high PMI).

### 4.2 3-way interactions

Figures 3 and 4 show these results as 3-way interactions. Results shown in Figure 3 simply confirm the general pattern found in Figure 1, modulated by frequency: the more similar two nouns

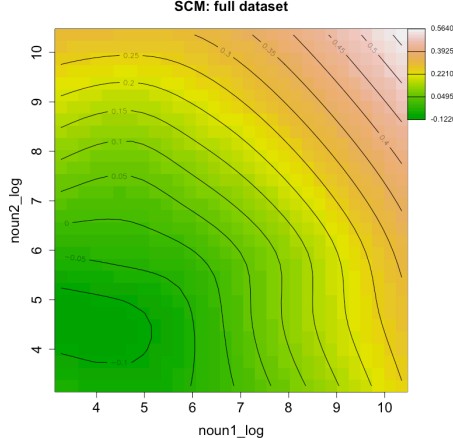

Figure 2: 2-way interaction and frequency effects in full dataset: the more frequent two nouns are, the more likely they are to share classifiers.

are, the more likely they are to share classifiers, but co-occurrence likelihood dampens that effect; frequency however increases the likelihood of classifier sharing across the board ($F(15.96) = 27.54$, $p < 0.0001$).

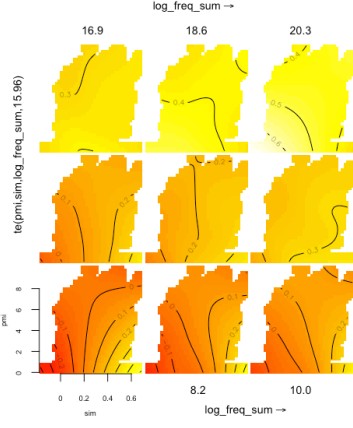

Figure 3: 3-way interaction between noun similarity, PMI, and summed noun log frequencies.

The interactions between individual log frequencies and increasing similarities shown in Figure 4 are more interesting. When taking into account 3-way interactions, we observe a pattern that could not have been seen when considering 2-way interactions only: in low similarity contexts, shared classifiers correlate with frequencies much like the patterns in the 2-way interaction illustrated in Figure 2; in high similarity contexts however, this tendency reverses: low frequency nouns are more likely to share classifiers, while high frequency nouns are more likely to appear with different sets

of classifiers. The high similarity context shows patterns that correspond to the frequency patterns reported for German by Dye et al..

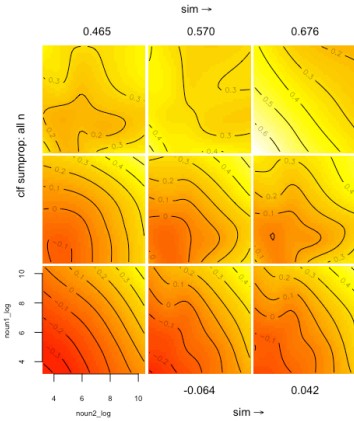

Figure 4: 3-way interaction between individual frequencies and noun similarity: In high similarity contexts, low frequency nouns are more likely to share classifiers, while high frequency nouns are more likely to appear with different sets of classifiers. In low similarity contexts the frequency effect is inversed. This three-way interaction is highly significant ($F(43.81) = 33.76$, $p < 0.0001$.)

### 4.3 Effects with different noun phrase structures

Interestingly, this effect is even stronger when only considering nouns immediately preceded by classifiers, as shown in Figure 5, but it disappears for nouns additionally combined with pre-nominal modifiers. As shown in Figure 6, in that scenario, frequency is the main effect significantly contributing to predicting shared classifiers (with more frequent nouns being more likely to share classifiers).

The effect of similarity in interaction with co-occurrence likelihood also differs across our two scenarios. Lower frequency nouns appear to share classifiers when they are more similar in the immediate classifier-noun sequence (Figure 7), but the pattern is reversed in noun phrases additionally containing pre-nominal modifiers (Figure 8). With increasing frequency, both pattern reverse and are still almost mirror images of each other in high frequency contexts. In both cases PMI contributes to mitigating similarity effects, but this interaction seems stronger in immediate classifier-noun sequences.

Overall, the GAM models reveal interesting 3-way interactions with significant differences de-

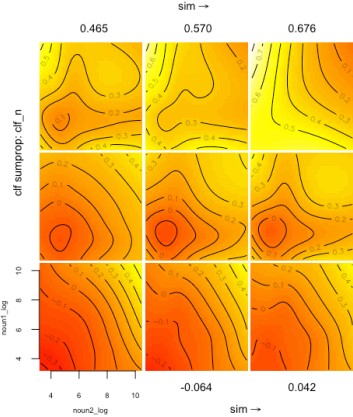

Figure 5: 3-way interactions in immediate classifier-noun sequences are significant ($F(41.44) = 16.67$, $p < 0.0001$) and their effects are the same as across all nouns with reversed patters for high vs. low similarity contexts.

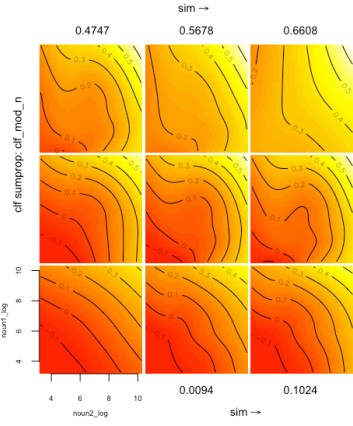

Figure 6: 3-way interactions in classifier-modifier(s)-noun sequences are significant ($F(32.77) = 26.93$, $p < 0.0001$) but higher frequency consistently correlates with shared classifiers, independently of similarity.

pending on noun phrase structure.

### 4.4 Additional effects

Interactions between overall noun frequencies and noun phrase structure further confirm what has been shown by the GAM models. Nouns immediately preceded by classifiers are significantly more frequent on average than nouns additionally modified ($F(1) = 3.69$, $p = 0.027$). Similarly, the mutual information between nouns and their immediately preceding classifiers is about 7.90 bits, while that between further modified nouns and their classifiers lies at 6.79. A one-way ANOVA confirmed that the difference in mutual information between

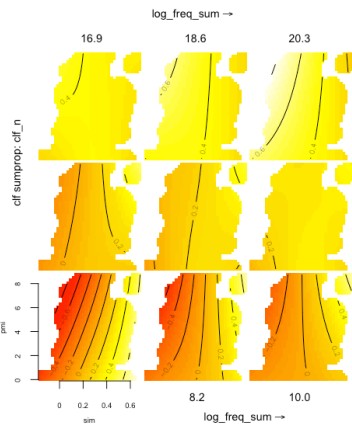

Figure 7: 3-way interactions between similarity, PMI and summed log frequencies for nouns immediately preceded by classifiers are significant ($F(27.49) = 28.28$, $p < 0.0001$).

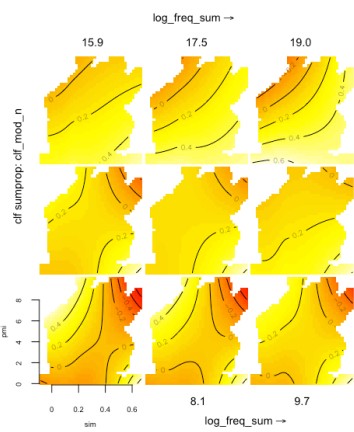

Figure 8: 3-way interactions between similarity, PMI and summed log frequencies for nouns preceded by additional pre-nominal modifiers are significant ($F(22.06) = 49.83$, $p < 0.0001$).

nouns and their immediately preceding classifiers and other nouns and their classifiers is significant ($F(1) = 29.41$, $p < 0.0001$).

## 5  Discussion and conclusion

Overall, our results show that noun-classifier combinations are sensitive to the nouns' frequencies and similarities. Nouns are overall significantly more likely to share the same classifiers if they are more frequent and/or more semantically similar. Co-occurrence likelihood (PMI) mitigates some of those effects.

Frequency, cosine similarity, and PMI measures capture different communicative scenarios (see description of the interactions between communica-

tive pressures in section 1). Frequency relates to speakers' familiarity with the data, the likelihood of whether they know the data or need to infer meaning from other data, and how much they rely on the data in communication. Cosine similarity as a measure of general semantic similarity corresponds to related meaning speakers can leverage when encountering a rare, previously unknown noun. PMI as a measure of co-occurrence likelihood relates to the likelihood that a speaker would have to differentiate between two nouns in context. We find that for words that tend to co-occur, there is an increase of likelihood of being combined with different classifiers even if two nouns are semantically similar according to their broader distributions. This effect is part of the communicative optimization we observe in the data. While two nouns that co-occur would likely be more semantically similar than two other random nouns, the fact that two nouns never closely co-occur does not mean that they do not otherwise share similarities in their distributions. In a later experiment, we increased the window size for the PMI measures to respectively 3, 5 and 10 word bidirectional windows. The effect was strongest with a 2 or 3-word bidirectional window and completely disappeared with 5 and 10 word windows. This further confirms that if the window is too large, co-occurrence is no longer a factor and cosine similarity and PMI no longer capture two different aspects.

What our results illustrate is that languages appear to optimize across frequencies to (i) enhance discriminability to facilitate efficient communication across the most widely used (i.e., most frequent) parts of the system, especially when words are likely to co-occur, and (ii) leverage regularity in the less frequent part of the system to maintain learnability of rarer words and structures potentially encountered late in life.

### 5.1  Relation to previous work

Our results on noun similarity effects are congruent with the cross-linguistic similarity effects reported by Williams et al. (2019). Interestingly, however, while their study used Mandarin as an example of a genderless language, our results prove that Mandarin does in fact make use of grammatical noun classification which is functionally very close to that of gender systems.

Our results on similarity and co-occurrence interactions further confirm results reported by Dye

et al. (2017a,b, 2018). When considering frequency however, our models show the opposite of the results reported by Dye et al. (2017a,b, 2018) for German. Interestingly, though, we do find the same results as Dye et al. when taking into account 3-way interactions – but only in high similarity contexts. There are several reasons why this might be the case:

(i) Mandarin and German noun classification could make use of opposing strategies when interacting with frequency. However, this interpretation does not seem likely given the more detailed understanding of the data obtained when considering 3-way interactions.

(ii) The sampling performed by the authors that only takes into account nouns appearing in all number and case combinations introduces a bias that goes against natural form distributions often described using Heap's law (Ross, 1960) and the way it applies to word form to lemma relations. According to Heap's law only a small subset of lemmas will ever be encountered in all their inflected forms in any given corpus, and increasing the corpus size to capture more complete paradigms comes with a diminishing return. Artificially restricting the sample of nouns to only nouns appearing as full paradigms in their corpus might have introduced a bias that the model could not control for.

(ii) While Dye et al. discuss possible 3-way interactions between frequency, similarity, and co-occurrence, they do not in fact compute any 3-way interactions. If they had taken into account these more complex interactions, they might have found patterns that mitigate their conclusion that frequency correlates with distribution over different gender classes. Our study is the first to use 3-way interactions between frequency, similarity, and co-occurrence between nouns to provide a complete understanding of their contextual intricacies related to patterns of noun class membership.

## 5.2 Novel findings

Interestingly, our first major finding is that, despite partial differences in reported results for 2-way interactions, the 3-way interactions we find confirm the conclusion presented by Dye et al. (2017b, 2018): noun class systems optimize to accommodate opposing communicative pressures that apply to high vs. low frequency nouns: while high frequency nouns are typically learned early and known by all, low frequency nouns may only be en-

countered late by native speakers, if at all. High frequency nouns thus mostly benefit from association with different classes that facilitate their discrimination in context, while low frequency nouns benefit from being associated with noun classes that can easily be extrapolated from the class semantically similar nouns belong to. These are exactly the pattern we see emerge for Mandarin classifiers when looking at 3-way interactions. Our study is the first full investigation of the this kind of 3-way interaction between frequency, similarity, and co-occurrence between nouns and their class membership.

Our second major finding confirms the idea that noun class systems can make use of grammatical and probabilistic noun classification simultaneously. In our results, we see that noun phrases that also contain additional pre-nominal modifiers, such as adjectives, do not display the inverting frequency effects depending on noun similarity described above.

Yet, when looking correlations between noun frequency and noun phrase structure and mutual information and noun phrase structure, we also find significant differences between nouns directly introduced by classifiers and nouns addionally smoothed by pre-nominal modifiers. We relate these differences to the existence of dual strategies in the noun classification system. High frequency nouns tend to appear in noun phrases that do not contain multiple additional pre-nominal modifiers that would probabilistically smoothe their entropy in context. They rely on grammatical noun classification operated by the classifiers directly. This is also reflected in the higher mutual information between such nouns and their classifiers. On the other hand, lower frequency nouns tend to be additionally smoothed by pre-nominally modifiers, share less mutual information with their classifiers and appear to rely on more probabilistic noun classification in the sense of Dye et al. (2018). For the first time, our study on Mandarin shows how differential behaviors displayed by class membership based on (a) noun pair similarity and (b) noun phrase structure reflects interactions between a grammatical and a probabilistic classification of nouns.

## 6 Limitations

A first limitation of this paper is that it does not initially replicate the studies its results are being

compared to. We attempted to reproduce the results in (Dye, 2017; Dye et al., 2018), but could not extract sufficiently precise information from those publications to re-run a version of their study. Our conclusions and discussion of their relation to Dye et al.'s previous findings are therefore based on an understanding of their general conclusions rather than concrete results.

Compared to many contemporary computational linguistic studies, the work presented in this paper was conducted on a relatively small data set. A larger study could be conducted by employing more powerful computational resources. In this paper, we instead decided to develop a controlled sampling technique which we applied to an originally large data set (3M sentences from The Leipzig Corpora Collection). Although reducing the amount of data fed to the models, this technique still preserves the frequency distributions of the full data set and has the advantage of being applicable with modest computational resources.

Finally, classifiers are commonly subcategorized into measure words and sortal classifiers. However, in a previous study (Wang and Walther, 2023), we showed that these two subcategories are functionally and distributionally sufficiently distinct to warrant treating them as separate syntactic (sub-)categories. In this study, we were most interested in sortal classifiers, but given their pre-nominal distribution, measure words were not irrelevant to the entropy reduction and noun classification questions we explore here. Preliminary experiments we conducted did not show any major differences between the two types of classifiers regarding the similarity, frequency, and co-occurrence measures presented in this paper. Part of the reason for this may be that even though measure words are typically described as more flexible in their combinations with different nouns, sortal classifiers also have a generic counterpart that can be used with most if not all nouns, which mitigates this bias. Nevertheless, the question of whether sortal and mensural classifiers play partly different roles remains interesting and would constitute an interesting extension of this study, that would take a closer look at the details of the classifier systems and not just at the functional aspects of noun classification. In (Wang and Walther, 2023), we already found that there was a significant difference between sortal and mensural classifiers in terms of entropy reduction of upcoming nouns: Sortal classifiers reduce entropy more than measure words.

## Acknowledgements

This project was supported by resources provided by the Office of Research Computing at George Mason University (URL: https://orc.gmu.edu) and funded in part by grants from the National Science Foundation (Awards Number 1625039 and 2018631).

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
