# OpenReview forum: "Mandarin classifier systems optimize to accommodate communicative pressures"
_EMNLP/2023/Conference — EMNLP 2023 Findings_

### Official Review · Reviewer_pvuj · 2023-07-31

**Soundness:** 3

**Excitement:**

2: Mediocre: This paper makes marginal contributions (vs non-contemporaneous work), so I would rather not see it in the conference.

**Paper Topic And Main Contributions:**

This paper analyzes 3-way interactions for nouns in Mandarin Chinese between frequency, similarity, and co-occurrence with nominal classifiers. Noun phrases are extracted from Leipzig Corpora Collection, and they use the fastText embeddings for similarity comparisons.
The paper looks into classifiers both immediately preceding and intervened by pre-nominal modifiers and shows that interactions of Mandarin nominal classifiers mirror grammatical and probabilistic nominal elements in other languages.

**Reasons To Accept:**

The analogy between nominal genders and classifiers and their interaction with nominal frequency and semantic similarity are theoretically motivating. Particularly, they observe opposite tendencies for frequent versus infrequent nouns.

**Reasons To Reject:**

Though linguistically provoking, ACL Rolling Review would not be a befitting venue for this paper. More data and/or modeling innovations would be necessary for an EMNLP long paper.

Some questions regarding the methodology:
- Since classifiers in Mandarin Chinese are more diverse than nominal genders but less than pre-nominal modifiers such as adjectives, the paper could build a more comprehensive analysis of this continuum by additionally comparing to the probabilistic end (pre-nominal modifier sequences) within Chinese and to other genderless languages such as English.
- Similarly, more background research on probabilistic pre-nominal modifiers would help understand the continuum. Particularly, classifiers, similar to adjectives, form a many-to-many (rather than one-to-many) correspondence to nouns.
- What are the reasons to downsample unique nouns from 44,620 to 390 (page 3)? Are those the nouns occurring "a minimum of 25 times"?
- There are more advanced Chinese dependency parsers than CoreNLP (Chen and Manning, 2014).

Several concepts or arguments require further explanation:
- What does "communicative pressure" refer to?
- It is unclear how ii and iii lead to the conclusion that "classifiers are inherently more semantically transparent" in line 167.
- Why is the shared membership score (SCM) an addition of 2 fractions (line 325)?
- The distinction between learning nouns earlier versus later is not introduced until page 8

**Reproducibility:**

3: Could reproduce the results with some difficulty. The settings of parameters are underspecified or subjectively determined; the training/evaluation data are not widely available.

**Reviewer Confidence:**

4: Quite sure. I tried to check the important points carefully. It's unlikely, though conceivable, that I missed something that should affect my ratings.

---

> ### Author Rebuttal · Authors · 2023-08-28
>
> *Thank you for taking the time to write a review on our paper! We hope that the additional information we provide below will help clarify the parts of our paper that weren’t sufficiently explicit.*
>
>
> ### Response to reasons to reject
> **_1. Though linguistically provoking, ACL Rolling Review would not be a befitting venue for this paper. More data and/or modeling innovations would be necessary for an EMNLP long paper._**
>
> This paper was submitted to the *Linguistic Theories, Cognitive Modeling and Psycholinguistics* track. The purpose of this study is to better understand the linguistic patterns of natural language and their relevance to human cognition. (It was not submitted to the *Language Modeling and Analysis of Language Models* track, where this criticism would potentially have been more relevant.)
>
> What we present is an empirical quantitative study of Mandarin for which we chose the data and methods necessary for obtaining significant results to our question: *are classifiers in Mandarin structuring nouns along noun classes that are optimized to accommodate cognitive pressures on language learning and processing?* The results we obtain with the amount of data we use are novel and significant. They show clear patterns that allow us to answer the question we ask. More data or more complex models were not necessary in this case.
>
> In addition, as long as results are not compromised in the process, there are ethical reasons for not running huge models when less processing power is sufficient and studies can be set up in such a way that they would be replicable by a larger community, including researchers who do not have access to expansive computing resources.
>
> **_2. Since classifiers in Mandarin Chinese are more diverse than nominal genders but less than pre-nominal modifiers such as adjectives, the paper could build a more comprehensive analysis of this continuum by additionally comparing to the probabilistic end (pre-nominal modifier sequences) within Chinese and to other genderless languages such as English._**
>
> This is partly what we are doing: we are showing that pre-nominal modifiers take over the categorizing function when they are present (see the difference between Figures 5 and 6). To the best of our knowledge, this is entirely new compared to previous studies and provides a really interesting insight into how systems self-organize and optimize for communication.
>
> Running experiments on English — or German for that matter —  would indeed be interesting, but this would be a different study and outside of the scope of this paper.
>
> The case of Mandarin is interesting in itself. It is particularly interesting because (i) Mandarin is generally considered genderless, and yet it presents noun classification patterns that mirror those of gendered languages when nouns are directly preceded by classifiers. (ii) We show that it combines grammatical and probabilistic noun classification, a type of complexity that would not be observable in a fully genderless language like English. (iii) Because classifiers are in a many-to-many relationship with their nouns, the system is in itself more complex than that of traditional gendered languages like German, which would likely present interactions between gender-based and adjective-based noun classification, but the gender system itself would be much simpler than a classifier system.
>
>
> **_3. Similarly, more background research on probabilistic pre-nominal modifiers would help understand the continuum. Particularly, classifiers, similar to adjectives, form a many-to-many (rather than one-to-many) correspondence to nouns._**
>
> That is something we explicitly mention lines 154-160, when explaining the problem, and again lines 310-312 when introducing our shared class membership (SCM) measure.
>
> **_4. What are the reasons to downsample unique nouns from 44,620 to 390 (page 3)? Are those the nouns occurring "a minimum of 25 times"?_**
>
> The nouns that were removed because they occurred less than 25 times were the 5,609 nouns that make up the difference between our original data set of 44,620 nouns and our final dataset of 39,011 nouns (approximately 12.5% of the data).
>
> From those 39,011 nouns, we selected 390 nouns (1% of the data). The nouns were randomly sampled across all frequency bands. This controlled sampling method was applied in order to preserve variation in frequency, as this frequency variation was part of the effects we wanted to explore in this study.
>
> We decided to conduct controlled sampling of our data rather than to run experiments over the whole dataset, so we would be able to obtain reliable results without requiring immense computing resources to run the experiments. Our goal was to design the experiments in  such a way that they would be easily replicable in the future, even by researchers who do not have access to high powered computing. (We also explain this in the paper’s limitation section.)
>
> The sampling code will be made publicly available with this paper if it is accepted.
>
> **_5. There are more advanced Chinese dependency parsers than CoreNLP (Chen and Manning, 2014)._**
>
> We apologize if we were not precise enough here. We did not use the 2014 version of the Stanford CoreNLP parser, but version 4.4.0 released in January 2022. The 2014 reference (Chen and Manning, 2014) is the reference that is meant to be cited when using any of the versions of CoreNLP, as indicated on the resource’s website.
>
> CoreNLP is a well-maintained resource. The very last version (4.5.4) was released in March 2023. CoreNLP also serves as a basis for multiple other state-of-the art parsers.
>
> **_6. What does "communicative pressure" refer to?_**
>
> We apologize if that concept was not explained enough at the beginning of the paper. We discuss this in lines 533-545, but we agree that we should introduce more details earlier on. We will dedicate more space to this in the introduction to make sure the paper is easier to read.
>
> “Communicative pressures” refer to competing pressures on language learning and language use, as defined by Blevins et al (2017): As speakers, humans face two main challenges: they have to *learn their language*, for which regularities across similar words (or structures) are the most optimal principle of organization; they also need to *use their language* in a way that allows them to efficiently communicate with others. For that, enhanced discriminability of words and structures provides the most optimal principle. Irregularities make words and structures more immediately recognizable/discriminable and allow messages to be processed more efficiently.
>
> As a result, a fully regular system would not be optimized for communication (one communicative pressure), whereas a fully irregular one would make a language extremely difficult if not impossible to learn (the other communicative pressure).
>
> **_7. It is unclear how ii and iii lead to the conclusion that "classifiers are inherently more semantically transparent" in line 167._**
>
> We mean to say that classifiers are more semantically transparent than gender markers.
>
> (i) A typical gender system like Spanish or German will have at most 2 or 3 genders, but there are hundreds of classifiers in Mandarin. As a consequence (and even though nouns can typically be introduced by several different classifiers), the combination of a noun with its (set of) classifier(s) is bound to be much more specific. The fact that individual classifiers are not combined with as many different nouns as individual gender markers makes their distributions, in other words their meanings, more specific.
>
> (ii) In addition to that, some tokens will be classifiers in some contexts and nouns in others and, as classifiers, they will retain part of the meaning they have as nouns. The fact that classifiers retain some of the meaning they have when used as a noun makes them semantically more transparent than gender markers.
>
> **_8. Why is the shared membership score (SCM) an addition of 2 fractions (line 325)?_**
>
> We want the SCM score to reflect how many classifiers are shared between two nouns, i.e., what proportion of the classifiers occurring with noun 1 also occur with noun 2 (size of the set of shared classifiers divided by the overall number of classifiers occurring with noun 1) **and** what proportion of the classifiers occurring with noun 2 also occur with noun 1 (size of the set of shared classifiers divided by the overall number of classifiers occurring with noun 2). The SCM score is the sum of those two proportions. Its values will be between 0 and 2.
>
>
> **_9. The distinction between learning nouns earlier versus later is not introduced until page 8_**
>
> We recognize that this should be explained more at the beginning of the paper and we will dedicate space to this in the introduction should the paper be accepted.

---

### Official Review · Reviewer_igeR · 2023-08-02

**Soundness:** 3

**Excitement:**

3: Ambivalent: It has merits (e.g., it reports state-of-the-art results, the idea is nice), but there are key weaknesses (e.g., it describes incremental work), and it can significantly benefit from another round of revision. However, I won't object to accepting it if my co-reviewers champion it.

**Paper Topic And Main Contributions:**

This paper aims to explain a functional purpose for the the classifier system in Mandarin. In Mandarin, quantifiers require an obligatory classifier which does not have a direct translation in English in most cases. For example, "two tables" is said as "two 张 tables"  where 张 is a classifier which is typically used for flat objects, e.g. tables, paper, beds, etc. The classifier is obligatory when quantifying unlike in English.

This paper argues that the classifier system allows for a reduction in entropy for upcoming nouns (i.e. having heard "two 张" you know that the upcoming noun must be something which is flat) and that communicative pressures have lead to this system. This is adapting previous work which argues that gendered languages reduce entropy (e.g. in French hearing "la" instead of "le" reduces the number of possible nouns to follow). Like gender, often the classifiers connection to a noun is seemingly arbitrary. Classifiers are somewhat more complex than gender as different classifiers can be used with the same noun, (although this is typically used for measure words, e.g. for "five rows of tables" in English, in Mandarin the phrase would be "five row_CL tables" where row_CL is a classifier meaning row.

The study looks at a corpus of 1M Mandarin sentences and evaluates the different effects of 1) co-occurence 2) similarity and 3) frequency on whether or not two nouns have an overlap in the classifiers they licence. Then, they fit a generalized additive model on a set a of noun-noun pairs to predict how classifier-overlap is predicted by the entaglement of 1), 2) and 3). For similarity, they use pretrained FastText embeddings and take their cosine similarity.

They find that high-similarity pairs, low frequency nouns will share classifiers more often than high frequency nouns. Conversely, if the similarity is low, it is high-frequency nouns that will share classifiers more often. They also find that the previous results on German gender systems by Dye et al. 2018, do not seem to occur with Mandarin classifiers (because frequency has an intermediating factor).



**Questions For The Authors:**

Question A) Why do you only sample 390 nouns from the total set of 39,011 distinct nouns? Furthermore why do you have two different numbers for the number of distinct nouns on line 247 and line 249 of page 3?

Question B) What would have constituted a negative result for this theory about classifiers? You get different results than Dye et al. do for German, but it is not clear whether this then means that the theory about classifiers is incorrect or not.

Question C) If classifiers, gendered articles and prenomial adjectives serve to reduce entropy about upcoming nouns, how do they work in head-initial languages such as Khmer where classifiers follow the noun? Would you expect a different pattern than in Mandarin?

**Reasons To Accept:**

- It provides an interesting account of a seemingly arbitrary part of natural language that might have a straightforward functional purpose.

**Reasons To Reject:**

- I wonder if the similarity measure is ideal as it certainly encodes information about the words' classifiers. Perhaps a better method would be to train novel embeddings on the corpus with classifiers erased.

- Much of the discussion is prose descriptions of the charts plotting the results. The work could be strengthened by elaborating more on why these results would or would not have been theoretically expected and what kind of predictions could be expected on the basis of these results for other languages. As it stands, the interpretation of the results remains rather difficult for the reader.

- Chinese classifiers come in two basic flavours, "count" and "mass". The former has no clear English analogue, e.g. we say five tables not five 张 tables. Mass classifiers do have a clear English equivalent, e.g. sān **bàng** ròu, "three **pounds** of meat" or yì **hé** dēngpào, "one **box** of lightbulbs". Mass classifiers do not really have the kind of arbitrary property the way gender systems do. I think it would be important to look at how the results look if one only looks at count-classifiers vs only mass-classifiers.

**Reproducibility:**

3: Could reproduce the results with some difficulty. The settings of parameters are underspecified or subjectively determined; the training/evaluation data are not widely available.

**Reviewer Confidence:**

3: Pretty sure, but there's a chance I missed something. Although I have a good feel for this area in general, I did not carefully check the paper's details, e.g., the math, experimental design, or novelty.

**Typos Grammar Style And Presentation Improvements:**

page 2, line 164: While "token" might make sense in a NLP context, it would probably be better to use lemma or word-form since you mean the a classifier might also function as a noun in different contexts rather than saying in some sentences, a single usage of a classifier has it also functioning as  a noun.

Table 1 page 3: this password**s**

The charts are a bit hard to read and could probably benefit from having their axes fully labeled. In particular, it took me some time to determine what  `clf sumprop clf_n` referred to.

---

> ### Author Rebuttal · Authors · 2023-08-28
>
> *Thank you for taking the time to write a detailed review of our paper, for giving us the opportunity to clarify some of its aspects and for raising interesting questions!*
>
> ### Preliminary clarifications
> **_1. You write: “Classifiers are somewhat more complex than gender as different classifiers can be used with the same noun, (although this is typically used for measure words, e.g. for "five rows of tables" in English, in Mandarin the phrase would be "five row_CL tables" where row_CL is a classifier meaning row.”_**
>
> This observation is absolutely true. However, when we mention that classifiers are more complex than gender systems, we don’t only mean that a noun can be introduced by more than one classifier at the same time, but most importantly that a given noun can combine with multiple different classifiers in different contexts. A gendered noun will always have the same gender regardless of the context it appears in.
>
> **_2. You write: “They also find that the previous results on German gender systems by Dye et al. 2018, do not seem to occur with Mandarin classifiers (because frequency has an intermediating factor).”_**
>
> Our results in the classifier-noun sequence scenario (Figure 5) are actually in line with what Dye et al. describe for German.
>
> ### Response to reasons to reject
>
> **_A. I wonder if the similarity measure is ideal as it certainly encodes information about the words' classifiers. Perhaps a better method would be to train novel embeddings on the corpus with classifiers erased._**
>
> This is true, but it would be more of a problem if all nouns always occurred with classifiers. We looked at the proportions of nouns occurring with and without classifiers in our data. This data is already heavily biased towards nouns with classifiers. (The data in our models only comprises nouns that occur with classifiers at least once. Those which never occur with classifiers were irrelevant to our experiments.)
>
> Among the nouns that are part of our data, only about 22% of their instances occur with a classifier (mean 22%; median as low as 7%). If we looked at all nouns, even those never occurring with classifiers, these numbers would be even lower. This means that noun embeddings will have been influenced by the classifiers, but only minimally. It is unlikely that another model would significantly change our overall findings.
>
> **_B. Much of the discussion is prose descriptions of the charts plotting the results. The work could be strengthened by elaborating more on why these results would or would not have been theoretically expected and what kind of predictions could be expected on the basis of these results for other languages. As it stands, the interpretation of the results remains rather difficult for the reader._**
>
> Thank you for pointing this out. We will do our best to streamline and emphasize the narrative more if the paper is accepted. In particular, we will dedicate part of the extra space to expanding the introduction to make the background and implications more understandable.
>
> **_C. Chinese classifiers come in two basic flavours, "count" and "mass". The former has no clear English analogue, e.g. we say five tables not five 张 tables. Mass classifiers do have a clear English equivalent, e.g. sān bàng ròu, "three pounds of meat" or yì hé dēngpào, "one box of lightbulbs". Mass classifiers do not really have the kind of arbitrary property the way gender systems do. I think it would be important to look at how the results look if one only looks at count-classifiers vs only mass-classifiers._**
>
> This is a very interesting observation, which we did not have the space to discuss in detail in the submitted version of this paper. Sortal and mensural classifiers are indeed different and there is abundance of linguistic literature trying to decide whether or not they should be considered one or two separate syntactic categories. We are most interested in sortal classifiers in this paper, but given their pre-nominal distribution, measure words are not irrelevant to the entropy reduction and noun classification questions we explore here.
>
> Preliminary experiments we conducted but didn’t include in this study for lack of space did not show any major differences between the two types of classifiers regarding the similarity, frequency, and co-occurrence measures presented here. Part of the reason for this may be that even though measure words are typically described as more flexible in their combinations with different nouns, sortal classifiers also have a generic counterpart that can be used with most if not all nouns, which mitigates this bias. This question however remains interesting and would be a great extension of this study, looking more into the details of the classifier systems and not just into the functional aspects of noun classification. For example, we already found that there was a significant difference between sortal and mensural classifiers in terms of entropy reduction for upcoming nouns. (Sortal classifiers reduce entropy more than measure words.)
>
> We will be happy to add some more information about this in the final version of this paper if it is accepted.
>
> ### Response to questions
>
> **_Question A) Why do you only sample 390 nouns from the total set of 39,011 distinct nouns? Furthermore why do you have two different numbers for the number of distinct nouns on line 247 and line 249 of page 3?_**
>
> The difference between the two numbers is due to the removal of 5,609 nouns (approximately 12.5% of the data) with frequencies below 25. We will make sure to better explain this point in the final version, as we agree that the two numbers are confusing without additional explanations.
>
> We sampled 390 nouns from our overall data partly for practical and partly for ethical reasons. Our goal was to run a model that does not require huge processing power without compromising on the reliability of the results. We sampled across frequency bands to have all scenarios represented. (We discuss this in the second point of the limitations section.) We will be including the code to allow the replication of the sampling procedure. We would like our findings to be replicable, even by researchers who do not have access to high powered computing.
>
> **_Question B) What would have constituted a negative result for this theory about classifiers? You get different results than Dye et al. do for German, but it is not clear whether this then means that the theory about classifiers is incorrect or not._**
>
> A negative result would have been the absence of any discernible pattern, or a radically different one from the ones described or predicted by Dye et al.
>
> We do get the same results as Dye et al. for nouns directly preceded by classifiers. But in addition to what they find, we also measure more complex system interactions, which quantitatively confirm their intuitions about probabilistic classification. The novelty here is that we show quantitative evidence of the interaction between grammatical and probabilistic noun classification. Our results illustrate that when probabilistic noun classification is available, noun classification does not require additional irregularities to discriminate high frequency high similarity nouns in context.
>
> **_Question C) If classifiers, gendered articles and prenomial adjectives serve to reduce entropy about upcoming nouns, how do they work in head-initial languages such as Khmer where classifiers follow the noun? Would you expect a different pattern than in Mandarin?_**
>
> Thank you for asking this very interesting question! We would definitely expect a different pattern for languages in which classifiers do not precede nouns. But we would need to measure the effects to be certain.
>
> That said, even in Mandarin, we have already found some promising results when looking at the rarer post-nominal classifiers. Post-nominal classifiers seem to be more likely to occur with nouns whose frequencies are higher. Our hypothesis is that they contribute to further specifying the meaning of more generic (high frequency) nouns in context. A study on other languages with more systematically used post-nominal classifiers would exceed the scope of this paper, but would constitute a great follow-up to this study!

---

### Official Review · Reviewer_DR8p · 2023-08-05

**Soundness:** 4

**Excitement:**

4: Strong: This paper deepens the understanding of some phenomenon or lowers the barriers to an existing research direction.

**Paper Topic And Main Contributions:**

The paper investigates nominal classification in Mandarin Chinese wrt use of classifiers. The work is contextualised from the perspective of the role of communicative efficiency as a guiding principle in language.

The key takeaway is that classifiers do in fact behave like gender as far as reducing the uncertainty of the upcoming noun is concerned. This is consistent with findings in German and English where the language relies on gender and pre-nominal modifiers to achieve the same.

The work also argues for a novel finding illustrated through the interaction between frequency and noun similarity. In particular, when the nouns are less similar, high frequency corresponds to more sharing of classifiers. However, when the nouns are similar, low frequency nouns are more likely to share classifiers compared to high frequency nouns.

Additionally, the above pattern was found to be contingent on the type of noun phrase. The above pattern was only present in nouns that were preceded by classifier and not in nouns which also had an adjectival modification. This is interpreted as the use of both grammatical as well as probabilistic ways of classifying nouns in a language.

**Questions For The Authors:**

A. In section 3.1 the authors state "We generated non-ordered noun pairs for those 1% and computed their summed frequency."; was this the measure reported in Figures 3 and 4? In other places only frequency is mentioned which is a bit confusing.

B. Isn't cosine similarity and PMI correlated?

C. What are these measures, i.e., summed frequency, cosine similarity and PMI, capturing about the data? They all seem to be capturing certain kind of similarity? e.g., co-occurences can also be deemed to capture semantic similarity (as in HAL, LSA); similarly if two words are highly frequent then they might also be similar?

D. "As can be seen from Figure 2, the likelihood of shared classifiers increases when nouns are more similar."; i suppose the authors meant Figure 1?

E. "the more similar two nouns are, the more likely they are to share classifiers, but co-occurrence likelihood dampens that effect" I wonder if this has something to do with the 2 window threshold for computing PMIs?

F. "The effect of similarity in interaction with co-occurrence likelihood ..." does co-occurrence mean PMI? please be consistent with terminology.

G. "The vectors were included as a quantifiable measure of a noun’s meaning based on the distributional hypothesis laid out by Harris (1954) and Firth (1957)." where was this used?

F. Taking together the findings of the 3-way interaction and the "correlations between noun frequency and noun phrase structure and mutual information and noun phrase structure", will it be reasonable to say that the low frequency effect (in high similarity situations) is being driven by noun phrases with pre-nominal modifiers? and the high frequency pattern (in the low similarity situations) is driven by noun phrases with classifiers? and if that is the case then the replication of Dye et al. is only applicable for the noun phrases with pre-nominal modifiers?

**Reasons To Accept:**

The paper discusses an important property in many languages, namely, nominal classifiers. This is perhaps the first attempt to systematically study the relation between classifiers and nominals from the perspective of communicative efficiency.

The paper attempts to generalise the various mechanisms that grammars employ to classify nouns. It relates the key results with previous findings on nominal classification based on gender. The paper nicely demonstrates that seemingly arbitrary grammatical features can in fact have functional underpinnings.

**Reasons To Reject:**

I thought that the rationale to include various metrics was not clearly mentioned. For example, while it is clear that the correspondence between nouns and classifiers should be influenced by nominal semantics. It is unclear why noun pair frequency and PMI should be relevant?

Relatedly, the 3 metrics, namely, frequency, cosine similarity, co-ocurrence/PMI can be thought as capturing some kind of similarity. In particular, co-ocurrence has been extensively used to determine semantic similarity. It is therefore unclear if cosine similarity and co-ocurrence could be correlated.

The above issues will have significant implication on the interpretation of the results.

**Reproducibility:**

4: Could mostly reproduce the results, but there may be some variation because of sample variance or minor variations in their interpretation of the protocol or method.

**Reviewer Confidence:**

4: Quite sure. I tried to check the important points carefully. It's unlikely, though conceivable, that I missed something that should affect my ratings.

**Typos Grammar Style And Presentation Improvements:**

- "Since knowing a noun’s classifier appears just as " --> appears to be

- "Yet, when looking correlations between ..." --> looking at

---

> ### Author Rebuttal · Authors · 2023-08-28
>
> *Thank you for taking the time to review our paper and for your thoughtful comments and questions!*
>
> ### Response to reasons to reject
>
> **_1. I thought that the rationale to include various metrics was not clearly mentioned. For example, while it is clear that the correspondence between nouns and classifiers should be influenced by nominal semantics. It is unclear why noun pair frequency and PMI should be relevant?_**
>
> Thank you for highlighting where our explanations were not entirely transparent, or came too late to be helpful.
>
> The reason why frequency and co-occurrences are expected to play a significant role in the organization of noun classes derives from their relation to competing communicative pressures as defined by Blevins et al. 2017. The explanations we give in the discussion are probably introduced too late and we should have added some more details as early as in the introduction. We will certainly do so in the final version of this paper should it be accepted. We are summarizing the interactions between communicative pressures below:
>
> **Regarding the role of frequency:**
> As speakers, humans are confronted with two main challenges: the first is *learning their language*, for which regularities across similar words (or structures) are the most optimal principle of organization; the second is *using their language* to efficiently communicate with others, for which enhanced discriminability of words and structures provides the most optimal principle. Irregularities make words and structures more item specific and thus increase their discriminability —  which allows for faster and more efficient communication.
>
> A fully regular system would not be optimized for communication, whereas a fully irregular one would make a language extremely difficult if not impossible to learn. However, not all parts of the system present the same challenge for learning: highly frequent words and structures are typically encountered early by learners and constantly reinforced. Once they are acquired, learning is no longer necessary. Irregularities only present a minor challenge towards the beginning of human language development, but they are of no problem at all later in life. They do however make communication faster and more efficient. This is especially useful for those words and structures that speakers use the most frequently.
> Infrequent words and structures, on the other hand, are typically only encountered late in life, if at all. Even proficient adult speakers will need to leverage their knowledge about related words and structures when they first encounter them. Regularity across similar structures is especially crucial in such cases. Since those words and structures are not as frequent, processing them faster is not as crucial to increasing overall communicative efficiency.
>
> The theoretical assumption behind those measures is therefore that languages optimize across frequencies to (i) enhance discriminability to facilitate efficient communication across the most widely used (= most frequent) parts of the system and (ii) leverage regularity in the less frequent part of the system to maintain learnability of rarer words and structures potentially encountered late in life.
>
> For noun classification, regularity comes from grouping together semantically similar nouns. In our case, a fully regular system would translate into semantically similar nouns always sharing the same classifiers. Irregularity on the other hand corresponds to distributing similar items over different classes. In our case, a fully irregular system would translate into semantically similar nouns never sharing the same classifiers.
> The hypothesis that we are testing here is whether Mandarin classifiers do indeed organize Mandarin nouns along those two competing principles. We find that semantically similar nouns indeed tend to share classifiers if they are infrequent and tend to be distributed across different classifiers when they are frequent. Speakers will overall expect highly frequent nouns to occur. In a given context, having a discriminable classifier will therefore be more efficient for reducing the entropy of the upcoming noun.
>
> **Regarding the role of co-occurrences:**
> Not all semantically similar words co-occur. For example, despite their very similar meaning, *gentleman* and *guy* are less likely to be used together, than *knife* and *fork*. Words that, in addition to being semantically similar, would also be likely to co-occur are the ones for which additional disambiguation would be functionally the most relevant. Introducing PMI as an additional measure allows us to test whether language systems are sensitive to co-occurrence likelihoods. In the case of Mandarin classifier systems, we find that this is indeed the case.
>
>
> **_2. Relatedly, the 3 metrics, namely, frequency, cosine similarity, co-occurrence/PMI can be thought as capturing some kind of similarity. In particular, co-occurrence has been extensively used to determine semantic similarity. It is therefore unclear if cosine similarity and co-occurrence could be correlated._**
>
> Cosine similarity evaluates semantic similarity across contexts that are broader than the ones we use for our PMI measures. The PMI measure looks at co-occurrences in a small bi-directional 2-word window and tests whether, in addition to general similarity, actual co-occurrences are reflected in system organization.
>
> ### Response to questions
>
> **_A. In section 3.1 the authors state "We generated non-ordered noun pairs for those 1% and computed their summed frequency."; was this the measure reported in Figures 3 and 4? In other places only frequency is mentioned which is a bit confusing._**
>
> Thank you for finding this imprecision in the text! The text only mentions frequency, but Figure 3 indeed reports results for interactions between cosine similarity and PMI modulated by the *summed* frequency of individual noun pairs (log), as indicated in the caption. Figure 4 shows the interactions based on *individual* noun frequencies (log) modulated by different degrees of cosine similarity. We will clarify this in the text.
>
> **_B. Isn't cosine similarity and PMI correlated?_**
>
> and **_C. What are these measures, i.e., summed frequency, cosine similarity and PMI, capturing about the data? They all seem to be capturing certain kind of similarity? e.g., co-occurences can also be deemed to capture semantic similarity (as in HAL, LSA); similarly if two words are highly frequent then they might also be similar?_**
>
> The measures are meant to capture different communicative scenarios (see description if the interactions between communicative pressures above). Frequency relates to speakers’ familiarity with the data, the likelihood of whether they know the data or need to infer meaning from other data, and how much they rely on the data in communication. Cosine similarity is used a measure of general semantic similarity. PMI relates to whether two nouns are likely to be used together.
>
> Two words could independently have fairly similar distributions without ever actually co-occurring with one another. The PMI measure specifically targets whether speakers will be likely to encounter two nouns closely together. We find that for words that tend to co-occur, there is an increase of likelihood of being combined with different classifiers even if two nouns are semantically similar according to their broader distributions. While two nouns that co-occur would likely be more semantically similar than two other random nouns, the fact that two nouns don’t ever closely co-occur doesn’t mean that they don’t otherwise share similarities in their distributions.
>
> **_D. "As can be seen from Figure 2, the likelihood of shared classifiers increases when nouns are more similar."; i suppose the authors meant Figure 1?_**
>
> Yes! Thank you for catching this!
>
> **_E. "the more similar two nouns are, the more likely they are to share classifiers, but co-occurrence likelihood dampens that effect" I wonder if this has something to do with the 2 window threshold for computing PMIs?_**
>
> Yes, since if they co-occur more, speakers will need to differentiate between them more often. This effect is part of the communicative optimization we observe in the data.
>
> In response to this question we also ran an additional experiment increasing the window size for the PMI measures to respectively 3, 5 and 10 word bidirectional windows. The effect is strongest with a 2 or 3-word bidirectional window and completely disappears with 5 and 10 word windows. This shows that if the window is too large, co-occurrence is no longer a factor and cosine similarity and PMI no longer capture two different aspects. We will include these additional results and corresponding figures in the paper.
>
> **_F. "The effect of similarity in interaction with co-occurrence likelihood ..." does co-occurrence mean PMI? please be consistent with terminology._**
>
> Yes, thank you! We will try to be more consistent in the final version of the paper to avoid lack of clarity on this matter.
>
> **_G. "The vectors were included as a quantifiable measure of a noun’s meaning based on the distributional hypothesis laid out by Harris (1954) and Firth (1957)." where was this used?_**
>
> The vectors refer to the fastText vectors we used to capture the nouns’ semantics. This is based on the distributional semantic hypothesis that distributions capture meaning, which goes back to those two references. (We hope we understood the question correctly and our answer clarifies it. We would be happy to specify further if there are any additional questions.)
>
> **_F. Taking together the findings of the 3-way interaction and the "correlations between noun frequency and noun phrase structure and mutual information and noun phrase structure", will it be reasonable to say that the low frequency effect (in high similarity situations) is being driven by noun phrases with pre-nominal modifiers? and the high frequency pattern (in the low similarity situations) is driven by noun phrases with classifiers? and if that is the case then the replication of Dye et al. is only applicable for the noun phrases with pre-nominal modifiers?_**
>
> We would think that it would be the other way around: the replication of Dye et al.'s claims on German only applies to nouns directly preceded by their classifiers. (This is being shown by the pattern inversion in Figure 5, which illustrates how semantically similar nouns will tend to combine with different classifiers the more frequent they are.)
>
> But in addition to that, our findings also provide quantitative evidence for another claim made by Dye et al., for which they however do not provide any measures — namely that probabilistic classification serves as an alternate classification device. Dye et al. compare grammatical and probabilistic noun classification across two languages with different gender profiles (German = 3 genders, English = no gender). What we show here is that this dual classification co-exists within one language and that there are interactions between the two, with measurable effects. In cases where nouns are only introduced by their classifiers, we observe the expected grammatical classification pattern inversion from Figure 5. In cases where nouns are also introduced by pre-nominal modifiers, there is no inversion. This shows that pre-nominal modifiers are an integral part of the system that helps with noun disambiguation, to the point that they 'disrupt’ the classifier-noun combination patterns found in cases without pre-nominal modifiers. When present, pre-nominal modifiers appear to serve as the primary classification device.  In such cases, the classifiers’ function is less crucial to the way information is managed and the whole system `adapts’ by avoiding the irregularity of the high-frequency/high-similarity scenario found in nouns occurring simple classifier-noun sequences.
>
> *Thank you again for giving us an opportunity to clarify the background, the role of the different measures, and the implications of this study. If this paper is accepted, we will dedicate part of the introduction to ensure all the connections are much more evident from the start. We will also include the additional experiments on different window sizes for PMI calculations and explain implications in greater detail in the Discussion section.*

---

### Meta-Review · Area_Chair_UbA2 · 2023-09-19

**Recommendation:** 3

**Metareview:**

> Summary

This paper investigates the classifier system in Mandarin Chinese, studying whether it shows signs of having been optimised for *communicative efficiency*.
Experimentally, they analyse pairs of nouns; studying how a pair’s frequency, semantic similarity, and co-occurrence probability (or PMI) predicts the amount of overlap in the nouns’ classifiers.
They find that, for low frequency nouns, more similar nouns will *more* likely share classifiers.
Similarly, they find that for high frequency nouns, more similar nouns will *less* likely share classifiers.
They take these results as sign of communicative efficiency.

> Meta review

Overall, the reviewers seem to find the paper’s research question interesting, and appreciate the connection this paper makes between gender and classifier systems. Further, this paper’s experiments seem well motivated. While some important parts of the paper were not clear (e.g., *what is the rationale behind the various used metrics?*, and *what exactly is the hypothesis being tested here, and what would constitute a negative result?*), I believe the author’s thorough responses answered those questions. In case this paper is accepted, I expect the authors will try to clear these sources of confusion in their manuscript as well.

The paper has a few drawbacks, however. In terms of scope, one of the reviewers suggested that the paper could be significantly strengthened by exploring both "count" and "mass" classifiers separately, and by comparing how these systems behave. Similarly, another review suggested that expanding these analyses to both adjectives and gender systems could provide a more comprehensive view of their role in communicative efficiency, especially since the authors argue for a similarity in the role of classifier and gender systems in natural languages.

Experimentally, one reviewer suggests that the used similarity metric here will contain information about the words' classifiers. The authors dismiss this concern by saying that only 22%/7% of the studied nouns’ instances occur with a classifier. Personally, I interpret these numbers the other way around, and believe that this is a large percentage of instances which could seriously impact the paper’s results. On the other hand, unless there is a systematic bias in how frequently nouns appear with classifiers (e.g., more frequent nouns appear with classifiers less often), I’d expect results to not change significantly since this bias will be present in most analysed nouns. In any case, training word2vec/fasttext is not particularly expensive, and I agree with reviewer igeR that running an experiment with novel embeddings trained on the corpus with classifiers erased would make these results more reliable.

On a related note, FastText uses subword information (I am not particularly familiar with how it processes Mandarin Chinese characters though). This means that your semantic similarity metric is likely to also encode wordform similarity as well. While this is not in itself an issue (as similar communicative efficiency arguments might be made for an "efficient" natural language to be optimised to distinguish wordform-similar words), it may change how these results should be interpreted. Rerunning experiments with, e.g., word2vec, could be interesting to analyse the extent to which that could have affected results.

---

### Decision · Program_Chairs · 2023-10-07

**Decision:**

Accept-Findings

**Comment:**

> Summary

This paper investigates the classifier system in Mandarin Chinese, studying whether it shows signs of having been optimised for *communicative efficiency*.
Experimentally, they analyse pairs of nouns; studying how a pair’s frequency, semantic similarity, and co-occurrence probability (or PMI) predicts the amount of overlap in the nouns’ classifiers.
They find that, for low frequency nouns, more similar nouns will *more* likely share classifiers.
Similarly, they find that for high frequency nouns, more similar nouns will *less* likely share classifiers.
They take these results as sign of communicative efficiency.

> Meta review

Overall, the reviewers seem to find the paper’s research question interesting, and appreciate the connection this paper makes between gender and classifier systems. Further, this paper’s experiments seem well motivated. While some important parts of the paper were not clear (e.g., *what is the rationale behind the various used metrics?*, and *what exactly is the hypothesis being tested here, and what would constitute a negative result?*), I believe the author’s thorough responses answered those questions. In case this paper is accepted, I expect the authors will try to clear these sources of confusion in their manuscript as well.

The paper has a few drawbacks, however. In terms of scope, one of the reviewers suggested that the paper could be significantly strengthened by exploring both "count" and "mass" classifiers separately, and by comparing how these systems behave. Similarly, another review suggested that expanding these analyses to both adjectives and gender systems could provide a more comprehensive view of their role in communicative efficiency, especially since the authors argue for a similarity in the role of classifier and gender systems in natural languages.

Experimentally, one reviewer suggests that the used similarity metric here will contain information about the words' classifiers. The authors dismiss this concern by saying that only 22%/7% of the studied nouns’ instances occur with a classifier. Personally, I interpret these numbers the other way around, and believe that this is a large percentage of instances which could seriously impact the paper’s results. On the other hand, unless there is a systematic bias in how frequently nouns appear with classifiers (e.g., more frequent nouns appear with classifiers less often), I’d expect results to not change significantly since this bias will be present in most analysed nouns. In any case, training word2vec/fasttext is not particularly expensive, and I agree with reviewer igeR that running an experiment with novel embeddings trained on the corpus with classifiers erased would make these results more reliable.

On a related note, FastText uses subword information (I am not particularly familiar with how it processes Mandarin Chinese characters though). This means that your semantic similarity metric is likely to also encode wordform similarity as well. While this is not in itself an issue (as similar communicative efficiency arguments might be made for an "efficient" natural language to be optimised to distinguish wordform-similar words), it may change how these results should be interpreted. Rerunning experiments with, e.g., word2vec, could be interesting to analyse the extent to which that could have affected results.